# The Spectrum of Adrenal Lesions in a Tertiary Referral Center

**DOI:** 10.3390/biomedicines12102214

**Published:** 2024-09-28

**Authors:** Carmen Sorina Martin, Marian Andrei, Bianca Alina Voicu, Miruna Alexandra Riță, Ana Alice Taralunga, Anca Elena Sîrbu, Luminita Nicoleta Cima, Iulia Stoian, Carmen Gabriela Barbu, Valentin Calu, Adrian Miron, Simona Fica

**Affiliations:** 1Department of Endocrinology, Diabetes Mellitus, Nutrition and Metabolic Disorders, Carol Davila University of Medicine and Pharmacy, 020021 Bucharest, Romania; bianca-alina.dumea@rez.umfcd.ro (B.A.V.); miruna-alexandra.ungureanu@rez.umfcd.ro (M.A.R.); anaalicetaralunga@gmail.com (A.A.T.); anca.sirbu@umfcd.ro (A.E.S.); luminita.cima@umfcd.ro (L.N.C.); iulia-simona.soare@umfcd.ro (I.S.); carmen.barbu@umfcd.ro (C.G.B.); simona.fica@umfcd.ro (S.F.); 2Department of Endocrinology, Diabetes Mellitus, Nutrition and Metabolic Disorders, Elias Emergency University Hospital, 011461 Bucharest, Romania; 3Cardiology Department, Bucharest Emergency University Hospital, 050098 Bucharest, Romania; 4Department of Radiology, ‘Dr. Carol Davila’ Central Military Emergency University Hospital, 010242 Bucharest, Romania; 5Department of Surgery, Carol Davila University of Medicine and Pharmacy, 020021 Bucharest, Romania; valentin.calu@umfcd.ro (V.C.); adrian.miron@umfcd.ro (A.M.); 6Department of Surgery, Elias Emergency University Hospital, 011461 Bucharest, Romania

**Keywords:** adrenal tumor, adrenal incidentaloma, adrenal cancer, adrenal paraganglioma, primary hyperaldosteronism, autonomous cortisol secretion

## Abstract

Background: Adrenal tumors are a common finding in clinical practice, and only detailed evaluation may reveal secretory and metabolic abnormalities or their malignant character. We aimed to highlight epidemiological data, rates of malignancy, clinical or secretory characteristics, and the cardiometabolic implications of adrenal masses. Methods: We conducted a retrospective analysis using data from the medical files of 474 patients with adrenal pathology hospitalized between January 2007 and January 2020, before the COVID-19 pandemic, using the ICD-10 codes. After applying inclusion and exclusion criteria, a total of 264 patients with adrenal tumors were enrolled in the study. Patients underwent clinical examination, abdominal imaging, and hormonal evaluation, and some of them underwent a pathological exam after adrenalectomy. Results: Median age at diagnosis was 56 (17) years, with 81.06% of patients being female. The median follow-up period was 41.5 (70) months, ranging from 6 months to 13 years. Adrenal tumors were most frequently seen in older female patients, with 83.47% of them being over 40 years old. The malignancy rate was 4.54%. Hormonally nonfunctioning tumors (71.95%) predominated, and overt hypercortisolism was present in 10.61% of patients, as was mild autonomous cortisol secretion in 5.31% of patients, primary hyperaldosteronism in 8.71% of patients, and adrenal paraganglioma in 3.41% of patients. Cardiometabolic comorbid conditions were similar in patients with functioning and nonfunctioning tumors. Conclusions: All patients with adrenal tumors should receive a complete hormonal workup and detailed malignancy risk assessment. Even though a hormonally active tumor predisposes to cardiometabolic comorbid conditions, a nonfunctioning lesion may also be associated with such disorders and needs thorough assessment.

## 1. Introduction

Adrenal glands, although small, are two endocrine glands which, through the hormones they secrete, play a crucial role in the proper functioning of the human body. The adrenal hormones help regulate metabolism, the immune system, blood pressure, response to stress, and other essential functions. Therefore, any sign of adrenal dysfunction requires special attention and thorough investigation whenever suspected.

Adrenal tumors, benign or malignant, represent a common finding, especially in recent years, since the technology of diagnostic methods has been greatly improved and is now widely used [1,2]. An adrenal tumor that is discovered while performing an imaging test for other reasons is called an incidentaloma. The clinical presentation of these patients may vary from no symptoms at all or nonspecific ones, to symptoms that may suggest abnormal hormone secretion or signs of malignancy. Therefore, the data on this topic may differ from one study to another, depending on their source—as the data may be clinical, surgical, imaging, or even autopsy series data—and on the country’s economic status [1].

Although it is now easier to find an adrenal tumor using imaging methods, an accurate diagnosis regarding the tumor type, whether it is benign or malignant, and the possibility of an associated abnormal hormone secretion often requires the intervention of a multidisciplinary team and many other investigations, such as hormonal tests, repeated imaging of different parts of the body, or even genetic testing. These aspects dictate the most appropriate management strategy, varying from only close surveillance to radical surgery [3,4].

Studies on the same subject have reported a greater prevalence of benign tumors, with the most frequent type being adrenal adenoma [5,6,7,8]. Regarding hormonal evaluation, in previous studies, between 69% and 85% of the tumors were nonfunctioning [3,4,5,9,10], keeping in mind that, usually, only around 15% of patients receive a complete biochemical workup [11]. Adrenal tumors are known to be more frequent in women, with a higher prevalence being shown among those between 40 and 70 years of age [12,13]. Also, adrenal tumors are usually unilateral findings, with some studies showing a higher frequency of left-sided ones [1,9]. As for tumor size, data available in the medical literature show that most tumors are less than 4 cm in diameter. The risk of malignancy is another concern, with malignant adrenal tumors being diagnosed in 8.6% of all patients. Although low, the malignancy rate correlates positively with tumor size [11]. An aspect worth mentioning is that, in a prior study, adrenocortical carcinomas (ACCs) were larger than adenomas, and most adrenal paragangliomas were larger than 4 cm [5,9]. As regards the association between nonfunctioning tumors and cardiometabolic disorders, there is a known relationship between these tumors and a higher prevalence of hypertension, hyperlipidemia, or diabetes mellitus, and these tumors are also associated with carotid intima-media thickness, epicardial adipose tissue thickness, and left ventricular mass, factors considered to be predictors of early cardiovascular disease [14,15,16].

The main purpose of this study was to carry out a retrospective review of the adrenal tumors managed in our tertiary referral center and to highlight the most relevant epidemiological, clinical, and paraclinical aspects of their pathology. Another aspect addressed in this study concerns the cardiometabolic implications of nonfunctioning adrenal tumors.

## 2. Materials and Methods

### 2.1. Patients and Study Protocol

We conducted a retrospective analysis using data from the medical files of 474 patients with adrenal pathology hospitalized between January 2007 and January 2020, before the COVID-19 pandemic, using the ICD-10 codes. Inclusion criteria were patients with adrenal tumors, admitted at least one time in our clinic, older than 18 years. Exclusion criteria were patients with insufficient data and other adrenal pathology, e.g., Cushing’s disease with adrenal hyperplasia, congenital adrenal hyperplasia, etc. Data collection and data entry were carried out by the same operator, and data were verified afterwards by a different person. After applying these criteria, a total of 264 patients with adrenal tumors were enrolled in the study, as seen in Figure 1.

Collected clinical and paraclinical data included age at diagnosis, gender, calculated body mass index (BMI) based on measured weight and height, systolic and diastolic blood pressure and heart rate, cardiovascular comorbid conditions, personal and family history, standard biochemical evaluation data (hemoglobin, platelet count, white blood cell (WBC) count, serum glucose, glycated hemoglobin, total cholesterol, LDL cholesterol, HDL cholesterol, triglycerides, serum creatinine, serum Na, and K), and specific hormonal measurements. Cortisol excess was assessed with 8 am plasma cortisol following a 1 mg dexamethasone suppression test (DST). A value of cortisol post-dexamethasone <1.8 µg/dL (50 nmol/L) excluded a cortisol-secreting tumor. In patients without clinical signs of overt Cushing’s syndrome, serum cortisol levels post-dexamethasone between 1.9 µg/dL (51 nmol/L) and 5 µg/dL (138 nmol/L) were considered as “mild autonomous cortisol secretion” (MACS), and values over 5 µg/dL (138 nmol/L) were taken as evidence for “autonomous cortisol secretion”. We also used other tests to confirm cortisol excess—24 h urine free cortisol (UFC), late-night salivary cortisol, 8 am and 11 pm plasma cortisol for circadian rhythm, low-dose and high-dose DST—as appropriate. Biochemical testing for adrenal paragangliomas was carried out using plasma-free metanephrines (N < 90 pg/mL or 456.3 pmol/L) and normetanephrines (N < 180 pg/mL or 982.8 pmol/L). Aldosterone excess was assessed by plasma aldosterone (ng/dL or pmol/L)/direct renin (µUI/mL) concentration ratio over 3.7, or 91 if converted to SI units. Other hormonal measurement parameters were 8 am plasma ACTH (N: 7.2–63.3 pg/mL or 1.58–13.93 pmol/L) and adrenal androgens when necessary. Plasma cortisol and ACTH were measured using the electrochemiluminescence immunoassay method, carried out using an ECLIA kit (Roche Diagnostics GmbH, Mannheim, Germany). For aldosterone or renin concentrations, chemiluminescent immunoassay technology was used, namely a CLIA kit (DiaSorin Inc., Saluggia, Italy), and for the quantitative determination of free metanephrine and normetanephrine in plasma, an enzyme immunoassay was carried out, using an ELISA kit (MP Biomedicals Germany GmbH, Eschwege, Germany). Based on hormonal evaluation, patients were further divided into two groups: 190 (71.97%) patients with non-secreting adrenal tumors and 74 (28.03%) with secreting adrenal tumors. Of those with secreting adrenal tumors, 28 (37.84%) had autonomous cortisol secretion, 14 (18.92%) had mild autonomous cortisol secretion, 23 (31.08%) had aldosterone secretion, and 9 (12.16%) had adrenal paraganglioma (of which one was malignant and one was atypical). Patients’ BMI, fasting glucose level, and blood pressure were classified according to international guidelines. Prediabetes was defined according to the latest ADA guideline criteria, based on fasting plasma glucose values between 100 and 125 mg/dL (5.6–6.9 mmol/L), HbA1c between 5.7 and 6.4% (39–47 mmol/mol), or 2 h plasma glucose during 75 g OGTT between 140 and 199 mg/dL (7.8–11 mmol/L) [17,18,19].

The diagnosis of adrenal tumor was established following structural imaging investigations (contrast-enhanced computed tomography (CT), MRI, etc.), from which data regarding the presence of unilateral or bilateral lesions; size; CT density, measured in Hounsfield units (HU); invasion in the neighboring structures and presence of necrosis; and hemorrhage or calcifications were obtained. A higher risk of malignancy was associated with a tumor >4 cm in size and >10 HU in density.

Data related to the treatment of our patients were recorded, including data pertaining to non-surgical follow-up, surgical treatment and the type of intervention (total or partial adrenalectomy), and medical therapy (if used). For some of those who underwent surgery, pathological data were available regarding the type and the benign or malignant character of the tumor. Tumors were classified based on the 2022 WHO classification of tumors of endocrine organs [20].

Adopting the data from a few smaller studies and the 2023 European guideline for adrenal incidentaloma, which mentions a higher likelihood of malignancy and more prevalent clinically significant hormone excess in individuals under 40 years old [4,5,21,22,23], we also analyzed our patients based on this age at diagnosis cutoff.

As necessary per protocol, patients were evaluated during first admission with complete hormonal workup and imaging procedures. Follow-up admissions were scheduled every 6 to 12 months, depending on the initial workup and diagnosis, in order to assess functional status and tumor growth.

### 2.2. Data Presentation and Statistical Analyses

The normality of data distribution was assessed using the Shapiro–Wilk test. Descriptive data are presented as means ± SD, as well as medians with interquartile range (IQR)/range or percentage. Between-group comparisons were carried out using parametric (independent sample *t*-test, *t*-test for paired samples, and a one-way analysis of variance (ANOVA) for more than two independent groups) or nonparametric (the Mann–Whitney U-test, a Kruskal–Wallis one-way ANOVA, and the Kolmogorov–Smirnov test) tests, as appropriate. A chi-squared test and Fisher’s exact test were used to compare proportions in large and small groups, respectively. Relations between continuous variables were analyzed using Pearson’s correlation parametric coefficient or Spearman’s rho nonparametric correlation coefficient. The SPSS statistical package for Windows, version 20.0. (IBM Corp. Released 2011. IBM SPSS Statistics for Windows, Version 20.0. Armonk, NY, USA: IBM Corp.), was used to perform all statistical analyses. A *p* value < 0.05 indicated statistical significance.

## 3. Results

### 3.1. Demographic, Clinical, and Cardiometabolic Characteristics of the Study Population

Of the 264 patients diagnosed with adrenal tumors included in our study, 214 (81.06%) were women, and 50 (18.94%) were men, with there being a median age at diagnosis of 56 (17) and a range between 18 and 89 years. The median BMI value was 29 (7.19) kg/m^2^, ranging from 17.60 to 60.23 kg/m^2^. One patient (0.40%) was underweight, 51 patients (20.56%) were in the healthy weight range, and the rest of them were either overweight (33.87%), class I obese (26.61%), class II obese (11.29%), or morbidly obese (7.26%). Only 47 of them (23.27%) presented with symptoms suggesting an adrenal tumor, while 155 (76.73%) were incidentally diagnosed via imaging tests conducted for other reasons. Other data, pertaining to associated comorbid conditions, are summarized in Table 1.

Of the 264 patients included in the study, 91 (34.47%) presented normal blood pressure values, while 44 of them (16.66%) had grade I hypertension, 59 (22.35%) had grade II hypertension, and 70 (26.52%) had hypertensive crisis or grade III hypertension. Overall, 31 (11.74%) patients presented with prediabetes, 58 (21.97%) presented with diabetes mellitus type 2 treated with oral hypoglycemic medications, and 4 (1.52%) had diabetes mellitus type 2 treated with insulin. The rest of the patients had normal glucose metabolism, and none of them had diabetes mellitus type 1. Abnormalities regarding lipid metabolism were present in 163 (61.74%) patients, 90 (34.09%) of which had mixed dyslipidemia, while 68 (25.76%) had hypercholesterolemia, and 5 (1.89%) had hypertriglyceridemia. In terms of cardiovascular comorbid conditions, 246 (93.18%) had a history of cardiometabolic disorders, of which 25 (9.47%) patients had valvular heart disease, 31 (12.6%) had coronary heart disease, and 12 (4.87%) had congestive heart failure. The cardiometabolic characteristics of the study population are presented in Figure 2. Extra-adrenal malignancy was observed in 59 (22.35%) patients.

### 3.2. Tumor Characteristics

Based on pathological reports, paraclinical data, and imaging data, all the tumors were classified according to the 2022 WHO classification of tumors of endocrine organs [20]. Most of the tumors (96.59%) were found in the adrenal cortex, while only 3.41% were located in the adrenal medulla. Out of the 255 tumors of the adrenal cortex, 195 (73.86%) were adenomas, 54 (20.45%) were cases of adrenocortical nodular disease, 3 (1.14%) were metastases of other malignant tumors, 2 (0.76%) were adrenocortical carcinomas (ACCs), and only 1 (0.38%) was an adrenal myelolipoma, as seen in Figure 3. According to the hormonal evaluation, out of the 195 adrenocortical adenomas, 148 (75.89%) were nonfunctioning tumors, 22 (11.29%) presented autonomous cortisol secretion, 9 (4.62%) presented MACS, and 16 (8.20%) secreted aldosterone. Out of the 54 adrenocortical nodular diseases, 36 (66.67%) were nonfunctioning, 6 (11.11%) secreted cortisol, 5 (9.26%) presented MACS, and 7 (12.96%) secreted aldosterone. The cases of adrenal myelolipoma, the metastases, and the ACCs were not associated with hormonal excess. All the other 9 tumors were located in the adrenal medulla and were adrenal paragangliomas, formerly known as pheochromocytomas. All adrenal paragangliomas presented catecholamine hypersecretion. Therefore, within the whole group of patients, 190 (71.97%) tumors were nonfunctioning, 28 (10.61%) secreted cortisol, 14 (5.30%) presented MACS, 23 (8.71%) secreted aldosterone, and 9 (3.41%) were adrenal paragangliomas that secreted catecholamines (Table 2).

The median tumor size, available only for 229 patients, was 2.40 (IQR = 1.62) cm, with values ranging from 0.4 cm to 9.80 cm. A total of 200 (87.34%) patients had tumors that measured less than 4 cm in the largest diameter, and only 29 (12.66%) patients had larger tumors.

Another aspect we considered was tumor density on CT scan, measured in Hounsfield units. Data were available for 73 patients, of which 65 (89.04%) had tumors with a density under 10 HU, and 8 (10.96%) had tumors with density over 10 HU.

Regarding the invasive nature of the tumors, only four of them (1.52%) appeared to invade other structures. Out of these, two were malignant (one ACC and one adrenal paraganglioma), and the other two were adenomas in close contact with inferior vena cava, without actually invading it.

The median follow-up period was 41.5 (70) months, ranging from 6 months to 13 years.

Follow-up imaging showed an adrenal tumor median growth of 0 (IQR = 1.35) cm, ranging from 0 (stable disease) to 10.50 cm, with 64.44% of adrenal tumors remaining stable.

Only 46 patients (17.42%) underwent surgery, while for the rest of them (82.58%), close surveillance was chosen.

As seen in Table 3, both adrenocortical tumors and adrenal paragangliomas were more frequent in women (81.06% and 77.78%, respectively). Regarding the median age at diagnosis, adrenal paragangliomas were diagnosed in significantly younger patients compared to adrenocortical tumors (40 (16) vs. 56 (18) years, *p* = 0.002), and they were significantly larger at diagnosis than the adrenal cortex tumors (3.85 (2.7) vs. 2.30 (1.53) cm, *p* = 0.003). There were significant differences between the patients with nonfunctional and functional tumors regarding the age at diagnosis, which was significantly lower in patients with functional tumors (49 (24) vs. 57 (14) years, *p* = 0.041), and tumor size at diagnosis, which was significantly higher in patients with functional tumors (2.9 (1.8) vs. 2.1 (1.48) cm, *p* = 0.016). With reference to tumor laterality and associated symptoms at diagnosis, functional tumors were more likely bilateral (*p* = 0.022) and symptomatic (*p* = 0.001).

### 3.3. The Characteristics of the Study Population Based on Gender

There was no significant difference regarding median age at diagnosis between men and women (55 (19) vs. 56 (17) years, *p* = 0.912).

No significant differences between males and females were reported regarding the prevalence of comorbidities, such as hypertension, diabetes mellitus, and dyslipidemia, and other extra-adrenal malignancies (*p* > 0.05), as seen in Table 4.

The most common tumors in women, as well as in men, were nonfunctional (71.50% and 72.00%, respectively), followed by cortisol-secreting tumors in women (11.21%) and aldosterone-secreting tumors in men (12.00%). The least common tumors, for women and men also, were adrenal paragangliomas (3.27% and 4%, respectively).

Tumor progression during follow-up (*p* = 0.493) and the prevalence of unilateral or bilateral tumors were similar between the two groups (*p* = 0.491).

### 3.4. The Characteristics of the Study Population Based on Age

We divided patients into two groups, patients under and over 40 years old at the time of diagnosis (15.53% and 84.47%, respectively); we compared the two groups, and the results are reported in Table 5. Grade II and III hypertension were significantly more frequent in patients over 40 years old (24.66% vs. 9.76% and 27.80% vs. 19.51%), while normal values of blood pressure and grade I hypertension were more frequent in patients under 40 years old (51.22% vs. 31.39% and 19.51% vs. 16.14%, *p* = 0.036). Prediabetes and diabetes mellitus type 2 were significantly more frequent in patients over 40 years old (12.56% vs. 7.32% and 26.46% vs. 7.32%, *p* = 0.006). Personal history of other types of tumors was a more common finding in patients over 40 years old (24.66% vs. 9.76%, *p* = 0.035).

### 3.5. The Characteristics of the Study Population Based on Tumor Size at Diagnosis

Additionally, we divided patients into two groups: patients with tumors larger than 4 cm and patients with tumors smaller than 4 cm in diameter. Of the 229 patients with available tumor sizes, 200 (87.34%) had tumors smaller than 4 cm, and only 29 (12.66%) had tumors larger than 4 cm in diameter. The percentage of male and female patients was similar between those with tumors over 4 cm in diameter and those with tumors under 4 cm in diameter (72.41% and 83.00% of females, respectively, and 27.59% and 17% of males, respectively) (*p* = 0.169). The median age at diagnosis was similar between the two groups (*p* = 0.998) and was 57 (18) years for patients with tumors under 4 cm and 54 (17) years for patients with tumors over 4 cm.

Most of the tumors larger than 4 cm, as well as the ones smaller than 4 cm, were incidentalomas and did not cause any specific symptoms. Symptoms suggesting the presence of an adrenal tumor were present in 24.14% of patients with tumors larger than 4 cm and in 18.50% of patients with tumors smaller than 4 cm. Associated type 2 diabetes mellitus was more frequent in patients with tumors larger than 4 cm (48.28% vs. 21.00%, *p* = 0.001). There were no significant differences between the two groups regarding other comorbidities, such as hypertension or extra-adrenal cancers (*p* > 0.05).

### 3.6. Cardiometabolic Implications of the Nonfunctioning Adrenal Tumors

To evaluate cardiometabolic implications, we compared the cardiometabolic comorbidities and cardiovascular parameters between patients with nonfunctional and functional tumors, and the results are reported in Table 6.

No significant differences between patients with nonfunctional and functional tumors were reported regarding the prevalence of cardiometabolic complications, apart from median systolic blood pressure, which was significantly higher in patients with functional tumors (140 (30) vs. 130 (20), *p* = 0.004).

## 4. Discussion

We performed a retrospective audit of 474 patients with adrenal pathology and 264 patients with adrenal tumors, managed in our tertiary referral center between January 2007 and January 2020, in order to highlight the most relevant epidemiological, clinical, and paraclinical aspects of their pathology and to analyze the cardiometabolic implications of nonfunctioning adrenal tumors. In our study, we included patients hospitalized before the COVID-19 outbreak because, during the pandemic years, many CT scans were conducted for SARS CoV2 infection, and more adrenal incidentalomas (AIs) may have been identified, as reported by de Magalhães et al. [23].

Adrenal tumors are rarely seen in young patients, and our study supports these data, with a median age at diagnosis of 56 (17) years, with 83.47% of patients being over 40 years old [1]. As previously reported in different studies, prevalence of adrenal tumor tends to increase with age [24,25].

As expected, most adrenal tumors were incidentally diagnosed and were adrenocortical adenomas less than 4 cm in diameter [11]. Regarding gender-oriented analysis, in our study, the percentage of female patients was significantly higher. Although Ahn et al., in their prospective study, reported that adrenal tumors were more frequent in men [9], other literature data also indicate a higher incidence of adrenal tumors in women [5,11,12].

Our findings highlight a lower malignancy rate than previously reported. We obtained a 4.54% malignancy rate, most of which (1.14%) were adrenal metastases. The second most frequent malignant tumor type was ACC (0.75%). Similar to our results, Ichijo et al. and Ebbehoj et al., despite reporting slightly higher malignancy rates—around 5.1% and 8.6%, respectively—found that most of the malignant tumors in their studies were also adrenal metastases [11,26]. Other authors have reported even higher malignancy rates, up to 10.3%, with the most frequent malignant tumor type being ACC [9,25]. This could be explained by referral bias in the reported studies.

Our results are in line with literature data regarding hormonal secretion, as 71.95% of tumors were found to be hormonally inactive. In large studies carried out previously on Italian or Polish patients with AIs, the percentage of hormonally inactive tumors was 85% and 83.1%, respectively [5,25]. The largest, most recent studies on American and Korean patients reported similar percentages of nonfunctional AIs (72.7% and 83.1%, respectively) [9,11].

In our study, the most frequently diagnosed overt hormone excess was hypercortisolism, with a frequency of 10.61%, which is higher than what previous studies reported (0.4–1.7%) [10,22]. Ichijoe et al. found a similar frequency of hypercortisolism (10.5%), but they used a cutoff of 3 μg/dL for cortisol after DST [26]. The reason for this discrepancy might be the different criteria used to define overt cortisol excess across different studies. Two studies that used almost the same criteria found hypercortisolism rates between 4.4 and 9.2% [5,9]. Regarding MACS, we found a frequency of 5.31% in our patients, which is lower than the 6.9% frequency reported by Ebbehoj et al., who used similar diagnostic criteria for MACS [11].

We found a frequency of 8.71% of aldosterone-producing tumors. Other frequencies reported in the literature vary between 1.6% and 6.1% [5,9,25]. Larger studies, including a higher number of patients, found a prevalence of aldosterone-producing tumors of 3.7% or 5.1%, respectively [11,26]. The higher percentage in our study may possibly be due to the lower number of patients included and the already preselected patients sent to our referral center.

Adrenal paragangliomas, formally known as pheochromocytomas, were found in 3.41% of the included patients, which were significantly younger and had larger lesions than those with adrenocortical tumors. Two of the most recent studies had similar results, with Ebbehoj et al. reporting a 1.1% incidence rate and Cyranska-Chyrek et al. reporting a 4.7% incidence rate, respectively [11,25]. Regarding age and tumor size at diagnosis, our results are similar to the results of Ahn et al., who reported a median age at diagnosis of 48 years and a median diameter of 3.8 cm at diagnosis [9].

Comparing nonfunctional with functional adrenal cortical tumors, we found that patients with nonfunctional tumors were significantly older and had smaller tumors that were more frequently unilateral, and as expected, few patients presented symptoms at diagnosis. One study, published by Öz et al., performed a similar evaluation between functional and nonfunctional adrenal tumors, comparing age, tumor size, and tumor lateralization. Its results, although similar, were not statistically significant [27].

Notably, many of our patients presented features of metabolic syndrome and cardiovascular comorbid conditions. There is growing evidence indicating that adrenal tumors, although nonfunctioning, are associated with factors increasing cardiometabolic risk, as reported by studies comparing healthy subjects with nonfunctional adrenal tumor patients [25,28]. In our study, we found no statistically significant difference when comparing the prevalence of cardiometabolic comorbid conditions between patients with functioning vs. nonfunctioning tumors, suggesting that the cardiometabolic impact of nonfunctioning adrenal tumors might be as important as for functioning ones.

Nonfunctioning adrenal incidentalomas with normal inhibition of cortisol secretion are not clinically silent; they more frequently present with endothelial dysfunction and early-stage cardiovascular remodeling, leading to cardiovascular and metabolic disorders. The exact pathogenic mechanisms linking these findings are not yet known, but subtle cortisol autonomy in adrenal adenomas, insulin resistance, and/or abnormal levels of inflammatory adipocytokines may play a pivotal role in the development and progression of atherosclerosis. There may be a bidirectional relationship between the presence of nonfunctioning adrenal tumors and insulin resistance. Insulin resistance may promote the growth of adrenal masses, and subtle but autonomous cortisol excess may lead to metabolic disturbances. Given the prevalence of adrenal incidentalomas, furthering out understanding of non-secreting adrenal tumors as an independent cardiovascular risk factor could have an important impact on public health policies, changing the paradigm in medical guidelines.

Concerning the limitations of this study, we acknowledge the limited originality of this topic. Our main objective was to perform an audit in our tertiary center before the COVID-19 pandemic, as data may vary around different centers, geographic regions, or time frames, and this may impact local everyday clinical practice. Owing to the retrospective nature of the study, another limitation is the significant number of patients excluded due to missing data. However, we limited the selection bias by systematically applying the same inclusion and exclusion criteria for all patients and by ensuring accurate data collection and data entry. Nevertheless, we would like to emphasize that, to the best of our knowledge, this is the first audit of adrenal tumors in a tertiary referral center in Romania, and it was caried out in an everyday clinical practice setting.

## 5. Conclusions

Even if most patients with adrenal tumors present with nonfunctioning adenomas less than 4 cm in diameter, all of them should receive a complete hormonal and imaging workup. A hormonally active tumor represents an extra reason for the development of features of metabolic syndrome or cardiovascular comorbid conditions, though there is evidence that a nonfunctioning adrenal tumor may also be associated with these comorbidities. Last but not least, the risk of malignancy should be evaluated for every adrenal tumor. Age, history of extra-adrenal malignancy, and imaging findings need to be carefully considered in order to establish the best management strategy for patients with adrenal tumors.

## Figures and Tables

**Figure 1 biomedicines-12-02214-f001:**
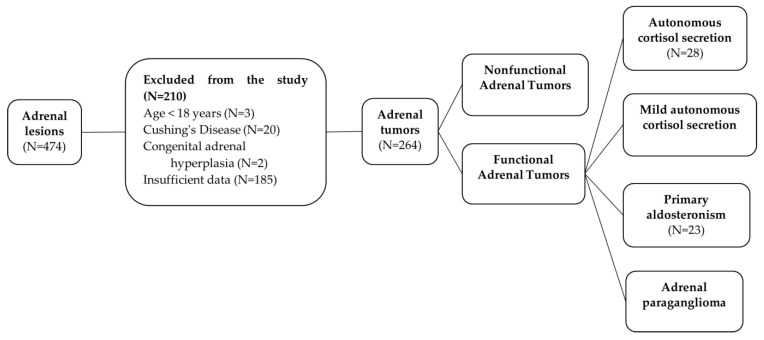
Patient recruitment flow chart.

**Figure 2 biomedicines-12-02214-f002:**
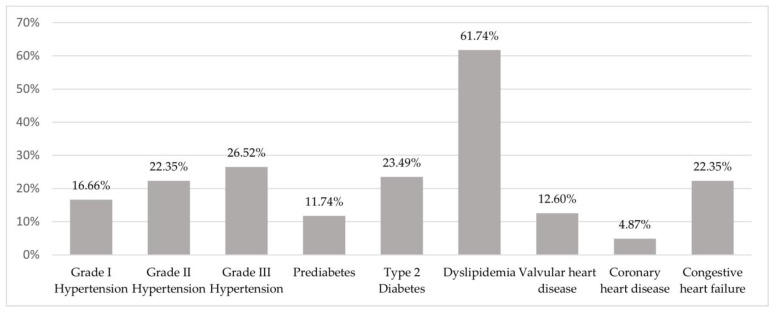
Cardiometabolic characteristics of the study population.

**Figure 3 biomedicines-12-02214-f003:**
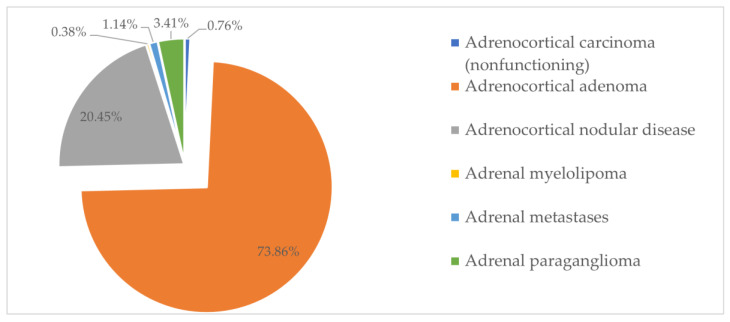
Types of adrenal tumors found in our cohort.

**Table 1 biomedicines-12-02214-t001:** Demographic, clinical, and cardiometabolic characteristics of the study population.

Parameter		Value
Age, median (IQR) (years)		56 (17)
Male/female, n (%)		50/214 (18.94/81.06)
BMI, median (IQR) (kg/m^2^)		29 (7.19)
Symptomatic tumor/incidentally discovered, n (%)		55 (20.83)/209 (79.17)
Associated comorbidities:		
Hypertension	Absent, n (%)	91 (34.47)
	Grade I, n (%)	44 (16.66)
	Grade II, n (%)	59 (22.35)
	Grade III, n (%)	70 (26.52)
Diabetes mellitus	Prediabetes, n (%)	31 (11.74)
	Type 2 Diabetes, n (%)	62 (23.49)
Dyslipidemia, n (%)		163 (61.74)
Cardiovascular comorbid conditions	Valvular heart disease, n (%)	31 (12.6)
	Coronary heart disease, n (%)	12 (4.87)
	Congestive heart failure, n (%)	59 (22.35)
Extra-adrenal malignancy, N (%)		59 (22.35)

**Table 2 biomedicines-12-02214-t002:** The types of adrenal tumors based on pathological reports, paraclinical data, and imaging data.

No.	Type of Tumor	No. of Cases	Percent (%)
I	Tumors of the adrenal cortex	255	96.59
1.	Adrenocortical carcinoma (nonfunctioning)	2	0.76
2.	Adrenocortical adenoma	195	73.86
	⁻ nonfunctioning	148	56.06
	⁻ autonomous cortisol secretion	22	8.33
	⁻ mild autonomous cortisol secretion	9	3.41
	⁻ aldosterone secretion	16	6.06
3.	Adrenocortical nodular disease	54	20.45
	⁻ nonfunctioning	36	13.63
	⁻ autonomous cortisol secretion	6	2.28
	⁻ mild autonomous cortisol secretion	5	1.9
	⁻ aldosterone secretion	7	2.65
4.	Adrenal myelolipoma	1	0.38
5.	Metastases	3	1.14
II	Tumors of the adrenal medulla	9	3.41
1.	Adrenal paraganglioma (pheochromocytomas)	9	3.41

**Table 3 biomedicines-12-02214-t003:** Characteristics of study population based on tumor location and hormonal evaluation.

Parameter		Total (n = 264)	Adrenal Cortex Tumors (n = 255)				*p*	Adrenal Medulla Tumors (n = 9)	*p* *
			Nonfunctional Tumors (n = 190)	MACS (n = 14)	Cortisol-Secreting Tumors (n = 28)	Aldosterone-Secreting Tumors (n = 23)			
**Gender, n (%)**	**F/M**	214 (81.06%)/50 (18.94%)	207 (81.18%)/48 (18.82%)					7 (77.78%)/ 2 (22.22%)	0.681
			153 (81.38%)/ 35 (18.62%)	54 (80.60%)/48 (18.82%)			0.88		
				13 (81.25%)/3 (18.75%)	24 (85.71%)/24 (85.71%)	17 (73.91%)/17 (73.91%)			
**Age, median (IQR) (years)**		56 (17)	56 (18)					40 (16)	**0.002**
			57 (14)	49 (24)			**0.041**		
				61 (13)	52.5 (26)	46 (15)			
**Tumor size, median (IQR) (cm)**		2.4 (1.62)	2.30 (1.53)					3.85 (2.7)	**0.003**
			2.1 (1.48)	2.9 (1.8)			**0.016**		
				3.25 (0.85)	3 (1.7)	1.6 (1.17)			
**Laterality, n (%)**	**U/B**	205 (77.65%)/ 59 (22.35%)	196 (76.86%)/59 (23.14%)					9 (100%)/0 (0%)	0.215
			153 (81.38%)/ 35 (18.62%)	52 (68.42%)/24 (31.58)			**0.022**		
				7 (43.75%)/9 (56.25%)	19 (67.86%)/9 (32.14%)	17 (73.91%)/6 (26.09%)			
**Symptomatic tumor, N (%)**		55 (20.83%)	27 (14.36%)	28 (36.84%)			**<0.001**		
**Extra-adrenal malignancy, N (%)**		59 (22.35%)	43 (22.87%)	16 (21.05%)			0.748		

*p* = *p* value between nonfunctional and functional adrenal cortex tumors; *p* * = *p* value between adrenal cortex and adrenal medulla tumors; F = female; M = male; U = unilateral adrenal tumor; B = bilateral adrenal tumor.

**Table 4 biomedicines-12-02214-t004:** The prevalence of different comorbidities and characteristics of the study population based on sex.

Parameter		Total (n = 264)	Female (n = 214)	Male (n = 50)	*p*
Hypertension	Normal values	91 (34.47%)	73 (34.11%)	18 (36.00%)	0.966
	Grade I	44 (16.67%)	35 (16.36%)	9 (18.00%)	
	Grade II	59 (22.35%)	49 (22.90%)	10 (20.00%)	
	Grade III	70 (26.52%)	57 (26.64%)	13 (26.00%)	
Diabetes mellitus	No diabetes	171 (64.77%)	140 (65.42%)	31 (62.00%)	0.294
	Prediabetes	31 (11.74%)	22 (10.28%)	9 (18.00%)	
	Type 2 diabetes	62 (23.49%)	52 (24.3%)	10 (20.00%)	
Dyslipidemia	Normal values	101 (38.26%)	85 (39.72%)	16 (32.00%)	0.312
	Dyslipidemia	163 (61.74%)	129 (60.28%)	34 (68.00%)	
Tumor size, median (IQR) (cm)		2.40 (1.62)	2.40 (1.6)	2.20 (1.81)	0.896
Laterality, n (%)	Unilateral	205 (77.65%)	168 (78.50%)	37 (74.00%)	0.491
	Bilateral	59 (22.35%)	46 (21.50)	13 (26.00%)	
Tumor progression, median (IQR) (cm)		0 (1.35)	0 (1.45)	0 (0.52)	0.493

**Table 5 biomedicines-12-02214-t005:** The prevalence of different comorbidities based on age.

Comorbidity		Total (n = 264)	Age ≤ 40 years (n = 41)	Age > 40 years (n = 223)	*p*
Hypertension	Normal values	91 (34.47%)	21 (51.22%)	70 (31.39%)	**0.036**
	Grade I	44 (16.67%)	8 (19.51%)	36 (16.14%)	
	Grade II	59 (22.35%)	4 (9.76%)	55 (24.66%)	
	Grade III	70 (26.52%)	8 (19.51%)	62 (27.80%)	
Diabetes mellitus	No diabetes	171 (64.77%)	35 (85.37%)	136 (60.99%)	**0.006**
	Prediabetes	31 (11.74%)	3 (7.32%)	28 (12.56%)	
	Type 2 diabetes	62 (23.49%)	3 (7.32%)	59 (26.46%)	
Dyslipidemia	Normal values	101 (38.26%)	16 (39.02%)	85 (38.12%)	0.912
	Dyslipidemia	163 (61.74%)	25 (60.98%)	138 (61.88%)	
Tumor size, median (IQR) (cm)		2.40 (1.62)	2.40 (1.6)	2.20 (1.81)	0.896
Extra-adrenal cancers	No	205 (77.65%)	37 (90.24%)	168 (75.34%)	**0.035**
	Yes	59 (22.35%)	4 (9.76%)	55 (24.66%)	

**Table 6 biomedicines-12-02214-t006:** Prevalence of cardiometabolic complications and cardiovascular parameters in patients with non-secreting tumors versus secreting tumors.

Parameter	Nonfunctional Tumors (N = 190)	Functional Tumors (N = 74)	*p*
Cardiometabolic complications, N (%)	173 (92.02%)	73 (96.05%)	0.239
High blood pressure, N (%)	109 (57.98%)	52 (68.42%)	0.115
Prediabetes, N (%)	25 (17.01%)	6 (10.91%)	0.284
Diabetes, N (%)	41 (21.81%)	21 (27.63%)	0.312
Dyslipidemia, N (%)	117 (62.23%)	46 (60.53%)	0.796
Hypercholesterolemia, N (%)	50 (26.60%)	18 (23.68%)	0.624
Hypertriglyceridemia, N (%)	5 (2.66%)	0 (0%)	0.326
Overweight, N (%)	58 (32.95%)	26 (36.11%)	0.634
Obesity, N (%)	85 (43.30%)	27 (37.50%)	0.121
Coronary heart disease, N (%)	19 (10.98%)	12 (16.44%)	0.239
Valvular heart disease, N (%)	16 (8.51)	9 (11.84)	0.403
BMI, mean ± SD (kg/m^2^)	29.60 (8.30)	28.27 (8.69)	0.117
Systolic blood pressure, median (IQR) (mmHg)	130 (20)	140 (30)	0.004
Diastolic blood pressure, median (IQR) (mmHg)	80 (20)	80 (20)	0.272

## Data Availability

Data supporting the published results are available from the corresponding author upon request.

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
