# Peer review of "The Spectrum of Adrenal Lesions in a Tertiary Referral Center"

_biomedicines, 2024, doi:10.3390/biomedicines12102214_

Round 1

Reviewer 1 Report

Comments and Suggestions for Authors

OVERALL ANALYSIS

The paper is a retrospective research article about the epidemiological data, rates of malignancy, clinical or secretory characteristics and cardiometabolic implications of adrenal masses in a tertiary referral center. The main streght is the good article organitation and presentation. The main limitation is the low originality of the topic.

SECTION BY SECTION ANALYSIS

TITLE AND ABSTRACT

I suggest to mention the statistical analysis in the methods and report some relevant patient's findings such as sex or age.

INTRODUCTION

Well written.

MATERIALS AND METHODS

The figure 1 (flow chart) is not clear. Only 264 had adrenal tumors? I suggest to start from all the patients with adrenal tumors and then exclude the patients with exclusion criteria. Regard this it is necessary to report in a clear way exclusion and inclusion criteria.

RESULTS

Well written.

DISCUSSIONS AND CONCLUSIOS

"There is growing evidence indicating that adrenal tumors, though nonfunctioning, are associated with factors increasing cardio-metabolic risk, as reported by studies comparing healthy subjects with nonfunctional adrenal tumor patients". Could you discuss this sentence? What are your hypothesis about this finding?

I suggest to add some relevant figures.

Reviewer 2 Report

Comments and Suggestions for Authors

Martin et al. have retrospective evaluated 264 adrenal tumors, of a selection of 474 cases were 210 were excluded mostly because of missing data, from a tertiary referral center in Romania. They find that 72% are non hormonal  functioning adenomas and that hypercortisolism is the most frequent hormonal aberration. Its an interesting finding that patients with AI have increased frequencies of the metabolic syndrome. For the radiological diagnosis size measurements was available in 229 patients and the density measurements with HU only available in 73 patients. Even thou many hormones are measured, the technique for laboratory measurements are missing.

Major concerns

The question of selection bias should be addressed thoroughly

Very few patients have measured HU witch is of paramount value in evaluating AI

Reference to the latest European guidelines are totally missing: Martin Fassnacht, Stylianos Tsagarakis, Massimo Terzolo, Antoine Tabarin, Anju Sahdev, John Newell-Price, Iris Pelsma, Ljiljana Marina, Kerstin Lorenz, Irina Bancos, Wiebke Arlt, Olaf M Dekkers, European Society of Endocrinology clinical practice guidelines on the management of adrenal incidentalomas, in collaboration with the European Network for the Study of Adrenal Tumors, European Journal of Endocrinology, Volume 189, Issue 1, July 2023, Pages G1–G42, https://doi.org/10.1093/ejendo/lvad066

For how long time were the patients followed? And how often?

Minor concerns

SI unites should be used or at least be shown throughout the manuscript

Metoxytyramin is not measured even thou they diagnose paraganglioma

What is the definition of prediabetes?

Table 3 is difficult to understand, what is U/B?

What kind of surveillance was done?

The paragraphs of sex and age could be shortened, there is not much to find here.

Why was the age of 40 years chosen?

In the discussion part, start with the main finding, the first paragraph in this section should be moved to the method part.

A translated copy of the informed consent should be added to supplemental material

Comments on the Quality of English Language

no comment

Round 2

Reviewer 2 Report

Comments and Suggestions for Authors

SI units or the possibilitiy to convert should be adressed:

example of SI units:

Blood sugar should be converted from  mg/dl to mmol/l  and HbA1c from % to mmol/mol  This apply for any measurements of hormones etc.

See: The international system of units (SI) - conversion factors for general use (govinfo.gov)
